# Electrical Coupling and Simulation of Monolithic 3D Logic Circuits and Static Random Access Memory

**DOI:** 10.3390/mi10100637

**Published:** 2019-09-23

**Authors:** Tae Jun Ahn, Bum Ho Choi, Sung Kyu Lim, Yun Seop Yu

**Affiliations:** 1Department of Electrical, Electronic and Control Engineering and IITC, Hankyong National University, 327 Jungang-ro, Anseong-si 17579, Gyenggi-do, Korea; jigo1235@hknu.ac.kr; 2Group for Nano-Photonics Convergence Technology, Korea Institute of Industrial Technology, Gwangju 500-480, Korea; bhchoi@kitech.re.kr; 3School of Electrical and Computer Engineering, Georgia Institute of Technology, Atlanta, GA 30308, USA; limsk@ece.gatech.edu

**Keywords:** circuit simulation, electrical coupling, monolithic 3D integrated circuit (IC), parameter extraction

## Abstract

In order to simulate a circuit by applying various logic circuits and full chip using the HSPICE model, which can consider electrical coupling proposed in the previous research, it is investigated whether additional electrical coupling other than electrical coupling by top and bottom layer exists. Additional electrical coupling were verified through device simulation and confirmed to be blocked by heavily doped source/drain. Comparing the HSPICE circuit simulation results using the newly proposed monolithic 3D NAND (M3DNAND) structure in the technology computer-aided design (TCAD) mixed-mode and monolithic 3D inverter (M3DINV) unit cell model was once more verified. It is possible to simulate various logic circuits using the previously proposed M3DINV unit cell model. We simulated the operation and performances of M3DNAND, M3DNOR, 2 × 1 multiplexer (MUX), D flip-flop (D-FF), and static random access memry (SRAM).

## 1. Introduction

Since the advent of Moore’s Law, semiconductor performance has been improved. However, silicon-based transistors below 10 nm have structural and physical fabrication challenges [1]. To break away from these problems, 3D structures have been researched as attractive solutions. Memory has already been mass-produced using 3D integration (3DI) in 3D NAND and 3D dynamic random access memory (DRAM) [2,3,4]. Various studies are also underway to use 3DI in logic circuits [5,6]. Monolithic 3DI (M3DI) is a method in which several layers are stacked step-wise and connected as nano-scale inter-layer vias. Compared to conventional 3D integrated-circuits (3DICs) based on through-silicon via (TSV), M3DI can reduce the length of wiring because peach can be nano-scale and can be partitioned at the gate level to improve IC performance (such as, delay, power consumption, device density, frequency, and bandwidth) without relying on scaling [7].

M3DI needs a limited thermal budget due to the sequential process on a single wafer. Although M3DI is limited to 2 h at 500 °C for performance stability of the bottom fully-depleted silicon-on insulator (FDSOI), it is possible to manufacture the M3DI without any degradation of performance through the research and development of the low-temperature process [8,9]. Recently, studies of 3D heterogeneous integration (e.g., complementary metal-oxide-semiconductor (CMOS) with nanoelectromechanical systems (NEMS), optical devices, or memory) that has been composed of CMOS and sensors or memory as a single stack have been reported [10,11,12]. In order to use M3DI in logic circuits, it is necessary to investigate electrical coupling between stacked and diagonally-stacked devices with inter-layer dielectric (ILD) and to enable circuit simulation considering electrical coupling. In previous studies, we investigated the electrical coupling of direct current (DC)/alterating current (AC) and transient device parameters in the monolithic 3D inverter (M3DINV) [13]. We proposed a new SPICE model to fully consider the investigated electrical coupling and extracted model parameters to create an M3DINV unit cell model for circuit simulation [14].

However, in order to design various CMOS logic circuits and perform circuit simulation using the proposed M3DINV unit cell model, it is necessary to investigate not only the electrical coupling between the top and bottom layers but also the additional electrical coupling in the diagonal direction by the adjacent transistors. It is necessary to investigate also the performances of various logic circuits and memories considering the electrical coupling.

In this paper, we propose the monolithic 3D NAND (M3DNAND) and monolithic 3D NOR (M3DNOR) gate structure (Section 2), and investigate the existence of additional electrical coupling diagonally by adjacent transistors (Section 3). Based on the electrical coupling investigated in Section 3, the circuit simulation for M3DNAND and M3DNOR is conducted using the M3DINV unit cell model to verify once more compared to the technology computer-aided design (TCAD) mixed-mode simulation results and to simulate various logic circuits and static random access memory (SRAM) using the M3DINV unit cell model. Finally, Section 5 concludes.

## 2. Structures

Figure 1 shows the cross-sectional views of M3DINV and M3DNOR. As shown in Figure 1a, M3DINV consists of N-type and P-type transistors in the top and bottom layers, respectively. M3DNOR consists of two M3DINV unit cells. The M3DINV unit cell has vertical electrical coupling, and M3DNOR may have electrical coupling in the diagonal direction (VEC-D) as well as vertical electrical coupling (VEC). Doping of MOSFET’s source/drain, LDD, and channel is 10^21^, 10^18^, and 10^15^ cm^−3^, respectively. The MOSFET was simulated at gate length (*L_g_*), gate oxide film (*T_ox_*), and ILD thickness (*T_ILD_*) at 30 nm, 1 nm, and 10 nm, respectively. SiO_2_ was used for gate oxide, ILD, and Box. The length of *L_gg_*, which is the distance between M3DINV unit cells, is 100 nm [13]. Figure 2 shows the proposed layouts and 3D structures of M3DNAND and M3DNOR.

## 3. Electrical Coupling

In this simulation, a device simulator ATLAS [15] of SILVACO was used. Table 1 shows the models, methods, and work functions used in TCAD simulation. The models used for device simulation are CVT, SRH, AUGER, and FERMI. The methods used for device simulation are NEWTON and GUMMEL. The gate work functions of the NMOSFET and PMOSFET are 4.57 and 4.9 eV, respectively.

Figure 3 shows the simulation results for the electrical coupling between N-type (A) and P-type (B) MOSFETs located in a diagonal direction in M3DNOR with *T_ILD_* = 10 nm. In order to investigate the vertical electrical coupling in the diagonal direction, the drain current-gate voltage (*I_ds_-V_gs_*) characteristics of top layer N-type (A) MOSFET in M3DNOR, as shown in Figure 2c, are simulated at different gate voltages (0 and 1 V) of diagonal bottom layer P-type (B) MOSFET. Here, the remaining electrodes except for the drain, gate, and source of N-type (A) MOSFET and the gate of P-type (B) MOSFET are fixed as 0 V. Symbols and solid lines are *I_ds_-V_gs_* characteristics of N-type (A) MOSFET when the gate voltages at P-type (B) MOSFET is 0 V and 1 V, respectively. In the proposed M3DNOR structure, it is confirmed that there is no electrical coupling because of the no shift of threshold voltage and no change of current.

Figure 4 shows the *I_ds_-V_gs_* characteristics of the top layer N-type (A) MOSFET in M3DNOR with *T_ILD_* = 10 nm at different *L_gg_*s. It is possible to verify that no electrical coupling exists when *L_gg_* is over 20 nm, but when *L_gg_* are below 20 nm, electrical coupling occurs. Because *L_gg_* is over 20 nm under the CMOS design rule of the channel length of 30 nm [16], the diagonal and vertical electric coupling can be ignored. In contrast to the top and bottom layers, because the heavily doped source/drain is located between the two N-type (A and B) MOSFETs, the electric coupling is blocked by the electrostatic shielding even though the distance is sufficiently influenced by electrical coupling. Simulation results show that the electrical coupling between the transistors located in the diagonal direction is blocked. Therefore, it is possible to simulate circuits for various monolithic 3D CMOS circuits, such as NAND and NOR, without additional parameter extraction using the M3DINV unit cell proposed in the previous study and external parasitic capacitances.

## 4. Simulation and Discussion

For circuit simulation, the LETI-UTSOI (version 2.1) model of HSPICE [17] was used for top and bottom transistors. Figure 5a is an equivalent model that can simulate a circuit considering the electrical coupling proposed in the previous study [13] and Figure 5b shows voltage transfer characteristics (VTCs) with the TCAD mixed-mode and HSPICE in M3DNAND at different *V_sub_*s (*V_dd_* and 0 V). HSPICE simulation results (lines) of the proposed model match those (symbols) of the TCAD mixed-mode accurately. This shows the validity of the HSPICE model. DC/AC and transient results were also verified [14] and used in the circuit simulation of various logic circuits in this study. Table 2 shows a summary of internal and external capacitances by metal lines (MLs) and monolithic inter-tier vias (MIVs) of M3DINV.

Figure 6 compares the VTCs and transient response simulation results of the TCAD mixed-mode and HSPICE of M3DNAND. Figure 6a shows the VTC characteristics of M3DNAND, and the symbols and solid lines denote the results of TCAD mixed-mode and HSPICE simulation, respectively. Black and red denote for input voltages B = 0 V and 1 V, respectively. Only when both inputs A and B are all “1,” the output is “0.” Figure 6b shows the transient response characteristics of M3DNAND. The symbols and solid lines denote the simulation results of TCAD mixed-mode and HSPICE simulation, respectively. Inputs A, B, and outputs are shown in order of one after the other from the top. Likewise, if both input A and B are all “1,” the output is “0,” and the internal capacitance is also well reflected.

Figure 7 compares the simulation result of VTC and transient response of TCAD mixed-mode and HSPICE of M3DNOR. Figure 6a is the VTC characteristic of M3DNAND and the symbols and solid lines denote the simulation results of the TCAD mixed-mode and HSPICE, respectively. Black and red denote for 0 and 1 V, respectively. It can be confirmed that the output is “1” only when both inputs A and B are all “0.” Figure 6b is the transient response characteristics of M3DNAND, and the symbols and solid lines denote the simulation results of the TCAD mixed-mode and HSPICE, respectively. Inputs A, B, and outputs are shown one after the other from the top. Similarly, it can be confirmed that the output is “0” only when both inputs A and B are all “0.” Based on the simulation results of the M3DNAND and M3DNOR, it was confirmed that the M3DINV unit cell model simulates well, reflecting the electrical coupling so that it can be used for circuit simulation of various logic circuits. However, M3DNAND and M3DNOR do not include extra capacitance and resistance due to monolithic inter-tier via (MIV) and metal line (ML). When circuit simulation of various logic circuits is performed, it is necessary to simulate by adding extra capacitance and resistance according to the structure of each circuit to obtain accurate results.

The M3DNAND and M3DNOR simulations, as shown in Figure 6 and Figure 7, confirmed that there was VEC without any additional electrical coupling diagonally. We used a M3DINV unit cell model to simulate various logics with the following two cases: One is *T*_ILD_ = 10 nm, which must consider VEC, and the other is *T*_ILD_ = 100 nm, which can neglect VEC. Figure 8a,b show the circuit simulation results of the 2 × 1 multiplexer (MUX) [18] and the D flip-flop (D-FF) [19] using the M3DINV unit cell model, respectively. Figure 8a shows inputs A and B, SLE, and outputs of 2 × 1 MUX in order of one after the other from the top. When *V_SEL_* = 0 V, *V_OUT_* = *V_B_*, and when *V_SEL_* = 1 V, *V_OUT_* = *V_B_*. It shows that 2 × 1 MUX operates well. The rising, falling propagation delays, and power consumption of 2 × 1 MUX (*T_ILD_* = 100 nm) without electrical coupling (black lines) are 2.5 ps, 1.4 ps, and 18.4 μW, respectively. The rising, falling propagation delays, and power consumption of 2 × 1 MUX (*T_ILD_* = 10 nm) with electrical coupling (red lines) are 2.9 ps, 1.7 ps, and 22.6 μW, respectively. Considering the electrical coupling, the rising and falling propagation delays, and power consumption are increased by 16%, 21.4%, and 22.8%, respectively. Figure 8b shows the clock, input D, output Q, and QB of D-FF in order of one after the other from the top. When the clock is rising edge, input D transfers to output Q and the inverted signal of input D transfer to output QB. It shows that D-FF operates well. The rising, falling propagation delays, and power consumption of D-FF (*T_ILD_* = 100 nm) without electrical coupling (black lines) are 12.3 ps, 4.4 ps, and 39.3 μW, respectively. The rising and falling propagation delays and power consumption of D-FF (*T_ILD_* = 10 nm) with electrical coupling (red lines) are 15 ps, 5.5 ps, and 41.1 μW, respectively. Considering the electrical coupling, the rising and falling propagation delays, and power consumption increased by 21.9%, 22.7%, and 4.6%, respectively.

Table 3 summarizes performance comparison of M3DI logics with/without electrical coupling. The capacitances, delays, and powers of the planar 2D CMOS and M3DIC structures without any electrical coupling (namely, *T*_ILD_ = 110 nm) are almost no difference with over 95% in the medium case (input slew = 37.5 ps), their areas are very different with over 40% [20]. Comparing the M3DI structures without electric coupling, which is similar to 2D CMOS except for cell areas, the static power and cell area of M3DI structure with electrical coupling were decreased and its dynamic power and average delay were increased.

Figure 9 shows the simulation results of the static noise margin (SNM) of 6 transistor (6T) SRAM [21] using M3DINV unit cell model. Figure 9a shows the SNM results in the retention operation, and Figure 9b shows the SNM results in the read operation. The retention SNMs of 6 T SRAM (*T_ILD_* = 100 nm) without electrical coupling for the high and low *V_out_* ranges are 275 and 237 mV, respectively, and their read SNMs are 101 and 88 mV, respectively. The retention SNMs of 6 T SRAM (*T_ILD_* = 10 nm) with electrical coupling for the high and low *V_out_* ranges are 281 and 265 mV, respectively, and their read SNMs are 121 and 107 mV, respectively. Considering the electrical coupling, the retention SNMs for the high and low *V_out_* ranges are increased by 2.2 and 11.8%, respectively, and the read SNMs for the high and low *V_out_* ranges are increased by 19.8 and 21.5% mV, respectively.

## 5. Conclusions

In this paper, we investigated the existence of additional electrical coupling in addition to the electrical coupling between top and bottom layers using the M3DNAND to simulate various logic circuits and memories using HSPICE models proposed in the previous study. As a result of the device simulation, it was confirmed that the electrical coupling by the diagonally-adjacent transistors of the newly proposed M3DNAND is blocked by the heavily doped source/drain. We verified that there is no additional electrical coupling, and the proposed M3DINV unit cell model was used to simulate the VTCs and transient response of the proposed M3DNAND and M3DNOR structures. Comparing the simulation results of TCAD mixed-mode to HSPICE using the proposed model, the validity of the proposed model was verified. The results of the VTC and transient response simulations are well matched so that the proposed model enables it to be used for circuit simulations of other logic circuits and memories. We simulated the operation and performances of M3DNAND, M3DNOR, 2 × 1 MUX, D-FF, and SRAM. Considering the electrical coupling, the rise and fall propagation delays, and power consumption of the 2x1 MUX increased by 16%, 21.4%, and 22.8%, respectively, and the rising, falling propagation delays, and power consumption of D-FF are increased by 21.9%, 22.7%, and 4.6%, respectively, and the retention SNM for the high and low *V_out_* ranges increased by 2.2% and 11.8%, respectively, and the read SNM for the high and low *V_out_* ranges increased 19.8% and 21.5%, respectively.

## Figures and Tables

**Figure 1 micromachines-10-00637-f001:**
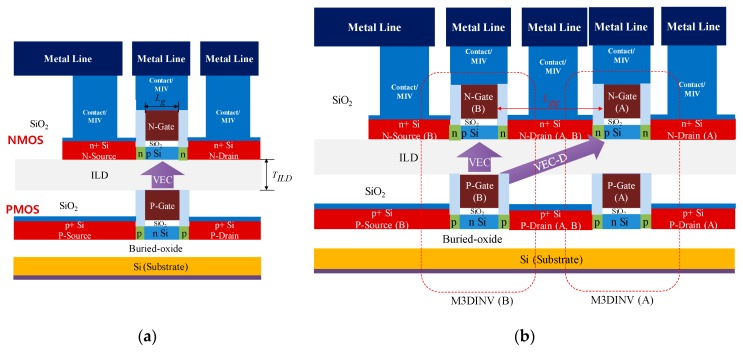
Schematics of two types of monolithic 3D integration (M3DI) cell structures. (**a**) monolithic 3D inverter (M3DINV) unit cell and (**b**) monolithic 3D NOR (M3DNOR) unit cell.

**Figure 2 micromachines-10-00637-f002:**
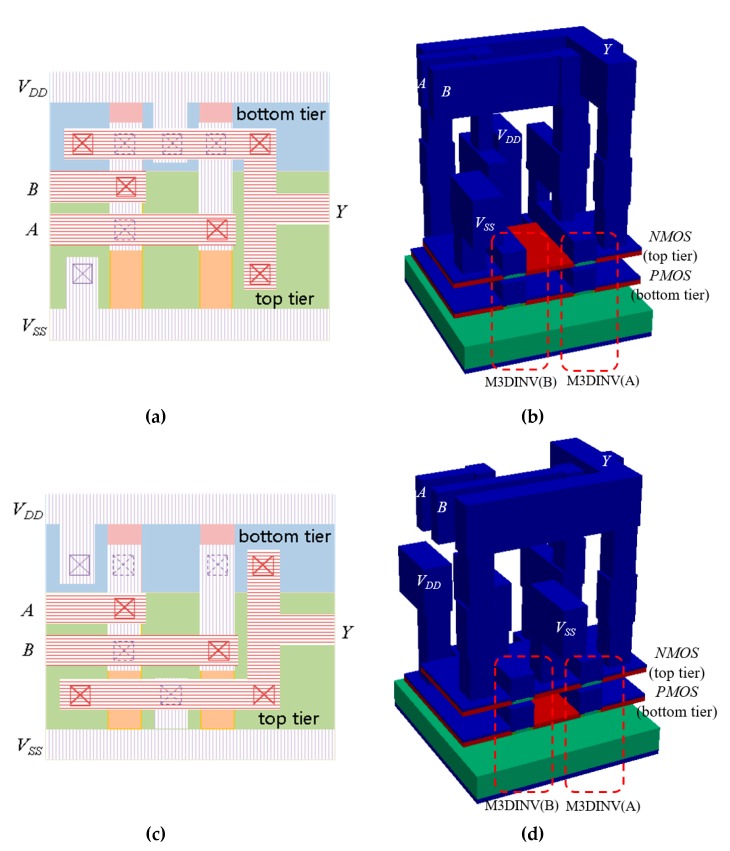
Two types of M3DI cell structure. (**a**) 2D layout of monolithic 3D NAND (M3DNAND), (**b**) 3D structure of M3DNAND, (**c**) 2D layout of M3DNOR, and (**d**) 3D structure of M3DNOR.

**Figure 3 micromachines-10-00637-f003:**
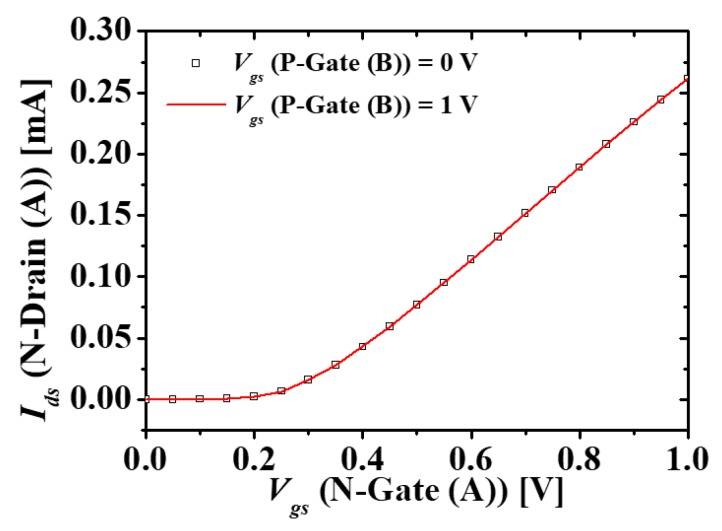
*I_ds_*-*V_gs_* characteristics of N-type (A) MOSFET when the gate voltage at P-type (B) MOSFET is 0 V and 1 V.

**Figure 4 micromachines-10-00637-f004:**
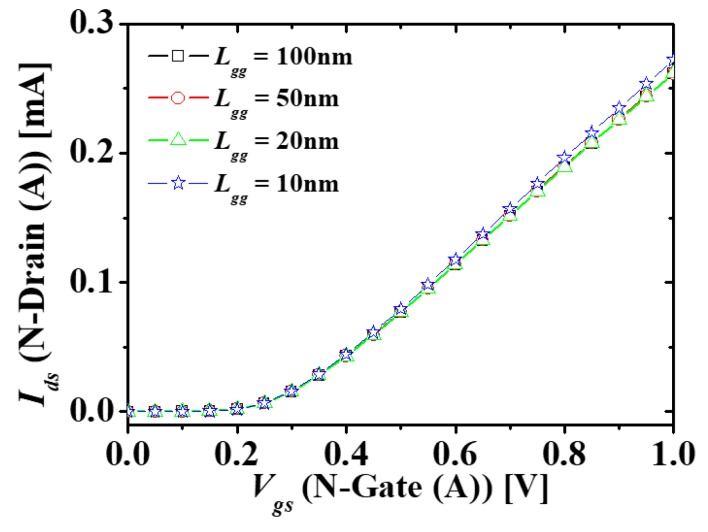
*I_ds_*-*V_gs_* characteristics of N-type (A) MOSFET in M3DNOR at different *L_gg_*s.

**Figure 5 micromachines-10-00637-f005:**
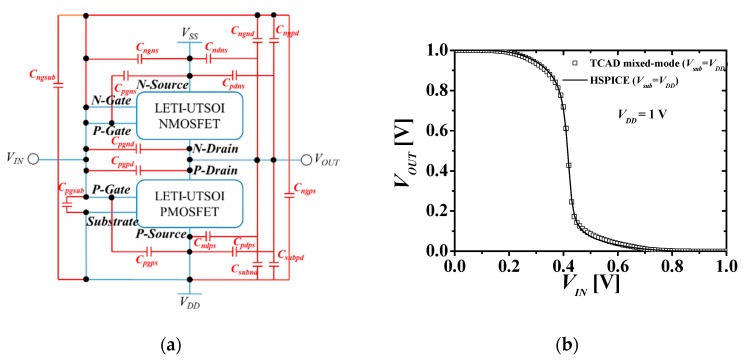
(**a**) Equivalent circuit of M3DINV [14] consisting of internal and external capacitances, where the capacitance caused by monolithic inter-tier vias (MIVs) and metal lines (MLs) are added. (**b**) Voltage transfer characteristics (VTC) of M3DINV [14]. *V_SS_* = 0 V and *V_DD_* = 1 V.

**Figure 6 micromachines-10-00637-f006:**
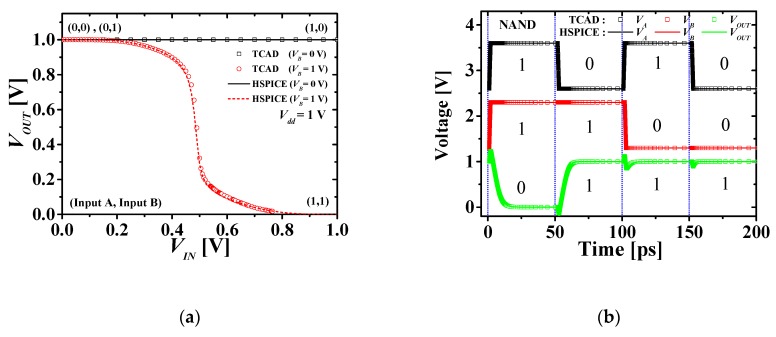
Comparison of (**a**) VTCs and (**b**) transient responses with TCAD mixed-mode and HSPICE of M3DNAND.

**Figure 7 micromachines-10-00637-f007:**
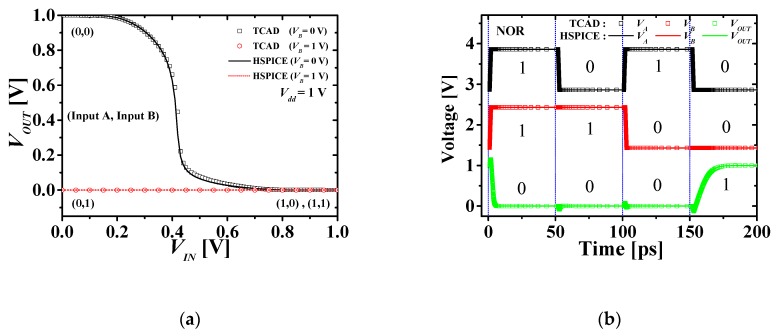
Comparison of (**a**) VTCs and (**b**) transient responses with TCAD and HSPICE in M3DNOR.

**Figure 8 micromachines-10-00637-f008:**
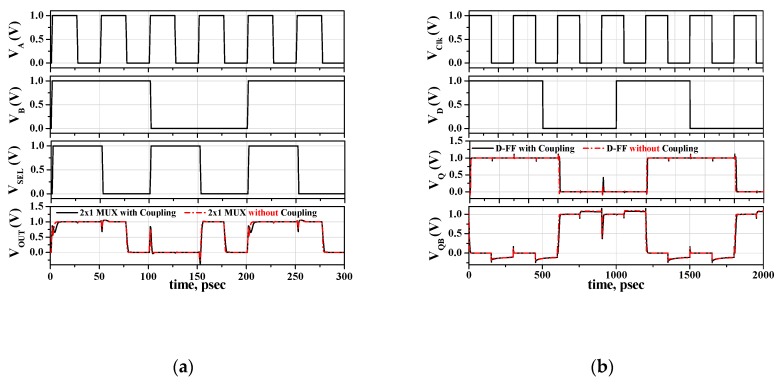
Transient response HSPICE simulation results of (**a**) 2 × 1 multiplexer and (**b**) D flip-flop.

**Figure 9 micromachines-10-00637-f009:**
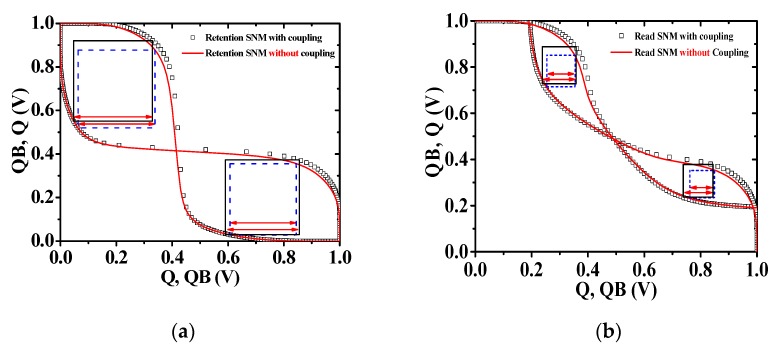
SNM HSPICE simulation results. (**a**) Retention operation SNM and (**b**) read operation SNM in 6T SRAM using M3DINV unit cell.

**Table 1 micromachines-10-00637-t001:** Description and dimension of models/parameters used in technology computer-aided design (TCAD) simulation.

Models/Parameters	Description	Value/Unit
CVT	Lombardi model and complete mobility model including doping density N, temperature T, and transverse electric field E_//_.	–
SRH	Shockley–Read–Hall recombination model	–
AUGER	Auger recombination model	–
FERMI	Fermi–Dirac carrier statistics	–
NEWTON	Newton method which solves a linearlized version of the entire nonlinear algebraic system	–
GUMMEL	Gummel method which solves a sequence of relatively small linear subproblems	–
*Φ_N_*	Gate work function of NMOSFET	4.57 eV
*Φ_P_*	Gate work function of PMOSFET	4.9 eV

**Table 2 micromachines-10-00637-t002:** Summary of internal and external capacitances.

Symbols	Description	Value (fF)
*C_ngns_*/*C_ngps_*	Top gate-top/bottom source capacitances of MOSFETs	0.0316/0.0006
*C_pgns_*/*C_pgps_*	Bottom gate-top/bottom source capacitances of MOSFETs	0.0007/0.08
*C_ngnd_*/*C_ngpd_*	Top gate-top/bottom drain capacitances of MOSFETs	0.0325/0.0018
*C_pgnd_*/*C_pgpd_*	Bottom gate-top/bottom drain capacitances of top/bottom MOSFETs	0.0007/0.08
*C_ndns_*/*C_ndps_*	Top drain-top/bottom source capacitances of MOSFETs	n.s./0.0001
*C_pdns_*/*C_pdps_*	Bottom drain-top/bottom source capacitances of MOSFETs	0.0002/n.s.
*C_ngsub_*/*C_pgsub_*	Gate-substrate capacitances of top/bottom MOSFETs	n.s./n.s.
*C_subnd_*/*C_subpd_*	Substrate-top/bottom drain capacitances of MOSFETs	n.s./0.0011

*n.s.: negligible small.

**Table 3 micromachines-10-00637-t003:** Performance comparison of M3DI logics with/without electrical coupling.

Performances	M3D with *T*_ILD_ = 100 nm (Neglecting VCE)					M3D with *T*_ILD_ = 10 nm (Including VCE)				
	INV	NAND	NOR	MUX	D-FF	INV	NAND	NOR	MUX	D-FF
Static power (nW)	5.1	3.97	5.17	7.1	42.2	4.89	1.63	2.41	4.21	17.4
						(−4.1%)	(−58%)	(−53.3%)	(−40.7%)	(−58.7%)
Dynamic power (μW)	8.55	12.4	12.2	18.4	39.3	9.85	13.9	13.8	22.6	41.9
						(15.2%)	(12%)	(13.1%)	(22.8%)	(6.6%)
Cell area (μm^2^)	0.043	0.087	0.087	0.013	0.261	0.043	0.087	0.087	0.13	0.261
	(0.101 *)	(0.217 *)	(0.217 *)	(0.986 *)	(2.117 *)	(−57.4%)	(−59.9%)	(−59.9%)	(−86%)	(−87%)
Average delay (ps)	3.875	4.925	4.64	1.95	8.35	4.93	5.45	5.22	2.3	10.25
						(27.2%)	(10.6%)	(12.5%)	(17.9%)	(22.7%)

* They mean the areas of 2D planar CMOS logics.

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
