# Peer review of "Electrical Coupling and Simulation of Monolithic 3D Logic Circuits and Static Random Access Memory"

_micromachines, 2019, doi:10.3390/mi10100637_

Round 1

Reviewer 1 Report

The manuscript is well written and verified. Moderate values exists in this study. I think it can be accepted. 

Author Response

Thanks for your kind comments.

Reviewer 2 Report

The ref. [13] [14] are much more sound and interesting than this manuscript. I didn't see the value for having a publication just verifying an inverter model to be applicable also on nor/nand cells.

Please elaborate on the 3-inverter structures of nor/nand gates, with detailed figures of merit covering area, delay and power, if they are assumed to be the novelty of this work.  

It's rather unclear for me what the electrical coupling in the mux and dff cells refer to. Please provide equivalent circuits of the devices with the parasitic capacitance that are shown critical in simulations highlighted.  

Author Response

The ref. [13] [14] are much more sound and interesting than this manuscript. I didn't see the value for having a publication just verifying an inverter model to be applicable also on nor/nand cells.

Please elaborate on the 3-inverter structures of nor/nand gates, with detailed figures of merit covering area, delay and power, if they are assumed to be the novelty of this work.

-> Thanks for your comments. I would revise the manuscript according to your comments. NAND and NOR can be composed of four transistors, two NMOSFETs and two PMOSFETs, respectively, and can be represented using two M3DINV structures. In the Ref. [13] and [14], we studied only the extraction of parameters for electrical coupling and circuit simulation of M3DINV structures. In this paper, we proposed an extension of M3DINV to use the M3DINV unit cell model without new parameter extraction for circuit simulation of logics consisting of more transistors. In order to use the M3DINV unit cell model, we investigated the electrical coupling in the diagonal direction which can be additionally structurally generated. It was confirmed that there is no electrical coupling in the diagonal direction in NOR and NAND gates. Based on this, the circuit simulation of NAND and NOR and various logics of M3DI structure was performed using M3DINV unit cell model.

-> Logic gates with over 3-inverters such as MUX and D-FF were also investigated in terms of delay, power, and area, as shown in Table 3.

It's rather unclear for me what the electrical coupling in the mux and dff cells refer to.

-> We agree your comments. The electrical coupling of MUX and D-FF means that the bottom layer affected electrically the top layer. In Ref. [13], the electrical coupling between the stacked transistors in M3DI structure changes as the thickness of the inter-layer dielectric (TILD) between the bottom and top layers changes. In this paper, with/without electrical coupling of MUX and D-FF means TILD = 10/100 nm, respectively. When TILD is 100 nm, there is no electrical coupling because the threshold voltage change of the top transistor between 0 and 1 V of bottom gate voltgae is less than 20 mV. In order to make you understand more easily it, we added a sentence related on electrical coupling in L.137-140, P.7.

Please provide equivalent circuits of the devices with the parasitic capacitance that are shown critical in simulations highlighted.

-> According to your suggestions, the equivalent circuit in Fig. 5 (a) is modified to the equivalent circuit with parasitic capacitance. We added Table 2 for a summary of internal + external capacitances by metal line (ML) and monolithic intertier vias (MIVs) of M3DINV in P.6.

Reviewer 3 Report

The manuscript (micromachines-576632) shows interesting results of electrical coupling and simulation of monolithic 3D logic circuits and SRAM. It reports a comprehensive analysis electrically. However, some outcomes still need to be addressed and verified before further confirm the results, shown as following:

In the introduction part, line 34-35, this referee is kind of surprise and confused about the FDSOI sentence, authors may need to give more background before further discuss in this part. Figure 3 (B) Figure is missing here.  For the simulation methods, can authors add a full Table of models (physical models), parameters (work function of metal), and conditions (etc. mesh) for referee and readers follow-up. It is very difficult to understand what would be the CVT, SRH, AUGER, and FERMI etc without any explain.  It would be great if authors can provide a simple benchmark of standard CMOS process (planar) with electrical coupling and without electrical coupling, as compared to current 3D structure with electrical coupling and without electrical coupling. What is the percentage change in dynamic and static power, effective area, switching speed, or delay in key metrics within different functional logic circuit (INV, NOR, NAND, MUX, and SRAM). These would be quite helpful for this referee and potential readers to catch the key impact contribution in this work analysis. 

Generally, due to the above comments, this referee would like to put the manuscript status as "Major Revision" in current phase. 

Author Response

The manuscript (micromachines-576632) shows interesting results of electrical coupling and simulation of monolithic 3D logic circuits and SRAM. It reports a comprehensive analysis electrically. However, some outcomes still need to be addressed and verified before further confirm the results, shown as following:

-> Thanks for your comments. I would revise the manuscript according to your comments.

In the introduction part, line 34-35, this referee is kind of surprise and confused about the FDSOI sentence, authors may need to give more background before further discuss in this part.

-> We agree your comments. According to your suggestions, we added the needs of the thermal budget for M3DI fabrication in L. 34-36, P.1.

Figure 3 (B) Figure is missing here.

-> (B) in the caption of Figure 3 does not mean Figure 3(B), but it means P-type (B) MOSFET.

For the simulation methods, can authors add a full Table of models (physical models), parameters (work function of metal), and conditions (etc. mesh) for referee and readers follow-up. It is very difficult to understand what would be the CVT, SRH, AUGER, and FERMI etc. without any explain.

-> We agree your comments. According to your suggestions, we added Table 1 for the simulation models and parameters at the top in P. 4 and described them in L.72-76, P.3.

It would be great if authors can provide a simple benchmark of standard CMOS process (planar) with electrical coupling and without electrical coupling, as compared to current 3D structure with electrical coupling and without electrical coupling.

-> We agree your comments. However, since standard CMOS is a planar structure, there is no electrical coupling between each transistor. We had to compare the planar CMOS with the M3DI with/without electrical coupling, but we were unable to add planar CMOS data due to time constraints (namely, paper revision requests within 8 days). However, in the Ref. [R1] shows that the capacitance and delay of the planar 2D CMOS and M3D IC structures with large ILD thickness are almost no difference. Therefore, instead of planar CMOS, power, capacitance, delay, and cell area of M3DI without electrical coupling are calculated, and the cell areas of planar CMOS structures are also calculted to compare with M3DI. We added L. 159-165, P.8 and Table 3 in P. 9.                                                                                                         

[R1] Lee, Y. J.; Limbrick, D.; Lim, S.K. Power benefit study for ultra-high density transistor-level monolithic 3D ICs. In Proceedings of 50th Annual Design Automation Conference, Austin, Texas, 29 May-07 June 2013; pp.1-19, doi: 10.1145/2463209.2488863.

What is the percentage change in dynamic and static power, effective area, switching speed, or delay in key metrics within different functional logic circuit (INV, NOR, NAND, MUX, and SRAM).

-> We agree your comments. According to your suggestions we added Table 3 for Performance comparison of planar CMOS process logics and M3DI logics in P. 8.

These would be quite helpful for this referee and potential readers to catch the key impact contribution in this work analysis.

Generally, due to the above comments, this referee would like to put the manuscript status as "Major Revision" in current phase.

-> Thanks for your comments.

Round 2

Reviewer 2 Report

Thanks for the updates! Most of my comments were addressed. The revision includes more detailed figures about M3D devices and some interesting comparisons with planer CMOS, which introduce values of this work for a publication.   

Minor typos on pp. 7, line 138: should be "as shown in Fig. 6 and Fig. 7" in my understanding.

Reviewer 3 Report

Authors have replied this referee comments in detail.